# Immunity from NK Cell Subsets Is Important for Vaccine-Mediated Protection in HPV+ Cancers

**DOI:** 10.3390/vaccines12020206

**Published:** 2024-02-17

**Authors:** Madison P. O’Hara, Ananta V. Yanamandra, K. Jagannadha Sastry

**Affiliations:** 1Department of Thoracic Head and Neck Medical Oncology, Division of Cancer Medicine, The University of Texas MD Anderson Cancer Center, Houston, TX 77030, USA; mpohara@mdanderson.org (M.P.O.); avyanamandra@mdanderson.org (A.V.Y.); 2UTHealth Graduate School of Biomedical Sciences, The University of Texas MD Anderson Cancer Center, Houston, TX 77030, USA

**Keywords:** natural killer cells, head and neck cancer, human papillomavirus, adjuvants, innate immunity

## Abstract

High-risk human papillomaviruses (HPVs) are associated with genital and oral cancers, and the incidence of HPV+ head and neck squamous cell cancers is fast increasing in the USA and worldwide. Survival rates for patients with locally advanced disease are poor after standard-of-care chemoradiation treatment. Identifying the antitumor host immune mediators important for treatment response and designing strategies to promote them are essential. We reported earlier that in a syngeneic immunocompetent preclinical HPV tumor mouse model, intranasal immunization with an HPV peptide therapeutic vaccine containing the combination of aGalCer and CpG-ODN adjuvants (TVAC) promoted clearance of HPV vaginal tumors via induction of a strong cytotoxic T cell response. However, TVAC was insufficient in the clearance of HPV oral tumors. To overcome this deficiency, we tested substituting aGalCer with a clinically relevant adjuvant QS21 (TVQC) and observed sustained, complete regression of over 70% of oral and 80% of vaginal HPV tumors. The TVQC-mediated protection in the oral tumor model correlated with not only strong total and HPV-antigen-specific CD8 T cells, but also natural killer dendritic cells (NKDCs), a novel subset of NK cells expressing the DC marker CD11c. Notably, we observed induction of significantly higher overall innate NK effector responses by TVQC relative to TVAC. Furthermore, in mice treated with TVQC, the frequencies of total and functional CD11c+ NK cell populations were significantly higher than the CD11c− subset, highlighting the importance of the contributions of NKDCs to the vaccine response. These results emphasize the importance of NK-mediated innate immune effector responses in total antitumor immunity to treat HPV+ cancers.

## 1. Introduction

High-risk human papillomaviruses (HPVs) are the underlying cause for over 5% of all cancers. Despite its decreasing incidence in developed countries, cervical cancer remains the second most common cause of death from cancer in women worldwide [1,2]. In addition, there has been an alarming increase in the incidence of HPV-associated oropharyngeal cancers (OPCs) in recent decades, relative to that of HPV+ cervical cancer, and now considered an epidemic [3,4]. Among fifteen different high-risk HPV genotypes, HPV-16 accounts for over 70% of cervical and 95% of HPV+ OPCs [1,4].

The viral-encoded oncoproteins E6 and E7, critical for the initiation and maintenance of HPV-associated malignancies, are ideal targets for therapeutic HPV vaccines [5,6]. However, vaccines targeting E6 and E7 have thus far shown modest efficacy in clinical trials despite inducing strong antigen-specific T cell responses [7]. The innate immunity forms the frontline defense against infections and diseases, including HPV-associated malignancies [8,9,10,11,12]. Therefore, vaccination strategies that significantly enhance antitumor immune responses with multiple effector functions to overcome the prevailing immunosuppressive tumor microenvironment are necessary to achieve maximum efficacy.

Adjuvants are an integral part of vaccine formulations that enhance the immunogenicity of vaccine antigens and activate innate immune mediators as well as effectors that bridge the innate and adaptive immune responses, specifically the antigen-presenting cells or dendritic cells [8,13]. The glycolipid α-galactosylceramide (aGalCer) is a potent agonist to natural killer T cells (NKTs), which are among the most effective innate immune modulators, for inducing the activation and maturation of dendritic cells (DCs) that, in turn, induce adaptive immune responses [14,15]. Toll-like receptor (TLR) ligand-based adjuvants are among the most commonly tested reagents for improving vaccine potency. Synthetic oligodeoxynucleotides containing one or more unmethylated CpG dinucleotides (CpG-ODN), which are common in bacterial and viral genomic DNA but not in vertebrates, serve as potent stimulators of TLR9 [16]. Unlike the TLR ligand-based adjuvants, QS21 is one of the best characterized natural saponin adjuvants [17,18].

We have previously reported that the HPV-16 E6 and E7 peptides are stimulators of T cell immune memory in HPV+ patients [19]; when formulated as a therapeutic vaccine with aGalCer and CpG-ODN adjuvants (TVAC) and delivered by the intranasal route, it induces strong mucosal and systemic antigen-specific T cell responses and protection against vaginal HPV tumors in C57BL/6 mice [20,21]. Here, we investigated the effectiveness of TVAC against an oral HPV tumor model and present evidence demonstrating the efficacy of a modified vaccine formulation where the aGalCer adjuvant in TVAC is replaced by another adjuvant QS21. The resulting vaccine, termed TVQC (therapeutic vaccine containing QS21 and CpG-ODN adjuvants), afforded protection in both the oral and vaginal preclinical HPV tumor models. We found that the curative efficacy of TVQC, in terms of >70% of mice exhibiting sustained long-term tumor regression, was associated with robust induction of multiple polyfunctional antitumor effector populations, in addition to HPV-antigen-specific CD8 T cells, including a subset of innate NK cells that express the dendritic cell marker CD11c, referred to as NKDCs.

## 2. Materials and Methods

### 2.1. Animals

C57BL/6J male and female mice (6–8 weeks) were procured from The Jackson Laboratory (Bar Harbor, ME, USA) and maintained in a pathogen-free environment. All animal studies were pre-approved and carried out in accordance with the University of Texas MD Anderson Cancer Center Institutional Animal Care and Use Committee (IACUC) guidelines. Mice were monitored daily for overall health and euthanized according to IACUC criteria.

### 2.2. Tumor Cell Lines

The mEER tumor cells are mouse tonsil epithelial cells expressing HPV-16 E6 and E7 along with H-ras, and they were a kind gift from Dr. John Lee (Avera Cancer Institute, Sioux Falls, SD, USA). These cells were maintained in complete Dulbecco’s Modified Eagle Medium (DMEM)/F12 media containing various supplements as previously described [22]; cells were sub-cultured at 80% confluence the day before tumor induction in mice. The TC-1-luc tumor cell line is of lung epithelial origin from C57BL/6 mice, expressing the E6 and E7 oncogenes of HPV-16 as well as firefly luciferase and H-ras. This cell line was a kind gift from Drs. T.-C. Wu and C. Hung (Johns Hopkins School of Medicine, Baltimore, MD, USA). The TC-1luc cells were maintained in complete Roswell Park Memorial Institute (RPMI) 1640 media supplemented with 10% heat-inactivated fetal bovine serum (FBS) (Atlanta Biologicals, Flowery Branch, GA, USA), 50 units/mL of penicillin–streptomycin and 50 μg/mL gentamycin.

### 2.3. In Vivo Tumor Challenge

Syngeneic male C57BL/6 mice were implanted with 4 × 10^4^ mEER cells in 50 μL phosphate-buffered saline (PBS) into the base of the tongue as described previously [23,24]. Mice were monitored twice weekly for body weight changes. Mice with oral tumors were euthanized when the mice lost 20% or more of their initial body weight. For the experiments involving the analysis of tumor-infiltrating leukocytes (TILs), 1 × 10^5^ mEER tumor cells in PBS were mixed in a 2:1 ratio with Matrigel (BD Biosciences, San Jose, CA, USA) for implantation. For intravaginal tumor challenge, 2 × 10^4^ TC-1 cells were implanted in the vaginal tract of diestrus-synchronized 6- to 8-week-old female C57BL/6 mice according to a previously described protocol [25,26]. Intravaginal TC-1 tumor growth was monitored using an In Vivo Imaging System (IVIS), a small animal imaging system (Perkin Elmer, Waltham, MA, USA), and expressed as the average luminescent signal in a selected region of interest (ROI) (*p*/s/cm^2^/sr) as a measure of tumor size. For the TIL experiments, 3 × 10^4^ TC-1 cells mixed 2:1 with Matrigel were used for intravaginal implantation.

### 2.4. Intranasal Vaccination

The therapeutic vaccine consisted of the following four peptides corresponding to the HPV-16 E6 and E7 oncoproteins: the E6_43–57_ peptide, Q15L (QLLRREVYDFAFRDL); the E6_49–58_ peptide, V10C (VYDFAFRDLC); the E7_44–62_ peptide, Q19D (QAEPDRAHVYNIVTFCCKCD); and the E7_49–57_ peptide, R9F (RAHVYNIVTF) [19,20]. Vaccine-grade (>90% pure) peptides were purchased from Elim Biopharma (Hayward, CA, USA) and used at 100 μg per dose of each, along with single or combinations of adjuvants (10 μg CpG-ODN, 5 μg QS-21, or 2 μg aGalCer). The R9F and V10C peptides represent murine H2b-restricted epitopes. Vaccine-grade ODN 1826 was purchased from InvivoGen (San Diego, CA, USA), and GMP-grade QS-21 was from Desert King (San Diego, CA, USA). ODN 1826 is a type B CpG-ODN specific for mouse TLR9 that is known to induce a Th1 response in addition to B cell activation, and it is a weak stimulator of IFNα and plasmacytoid DC [27,28]. The adjuvant aGalCer was purchased from DiagnoCline (Hackensack, NJ, USA). All adjuvants were reconstituted according to the manufacturers’ recommendations. Tumor-bearing mice were randomized, and on days 5 and 11 following tumor implantation, vaccines were administered via the intranasal route as described previously [23,26]. Untreated mice that received intranasal endotoxin-free PBS or those treated with a mixture of peptides and individual adjuvants served as control groups.

### 2.5. Magnetic Resonance Imaging (MRI)

For the mEER oral tumor model, tumor volume was measured using MRI. Mice were imaged under 2% isoflurane vapor anesthesia in a 7T small animal MR scanner (Biospec, Bruker Biospin Inc., Billerica, MA, USA) using transmit/receive volume coils with an inner diameter of 35 mm. The following parameters and settings were used for acquiring image sequences: field-of-view 4 × 3 cm^2^, matrix 256 × 192 and spatial resolution 156 microns; a T2-weighted coronal Rapid Acquisition with Relaxation Enhancement (RARE) with T2-Sagital; echo time 38 ms; repetition 1800 ms; slice thickness 0.50 m; and slice gap 0.50 mm. Tumor volume was calculated based on the image sequences analysis in three dimensions using Image J software version 1.53 (NIH, Bethesda, MD, USA) after defining the region of interest (tumor) on all possible sections.

### 2.6. Immune Cell Isolation

On day 15 or 16 post-tumor challenge, mice were euthanized and tumors, spleens, and tumor-draining lymph nodes (LNs) were collected to characterize the cell-mediated antitumor immune responses. Briefly, tumors were digested in complete RPMI media containing Collagenase H (Sigma-Aldrich, St. Louis, MO, USA) and DNase (Roche, Indianapolis, IN, USA), and they were incubated at 37 °C for 45 min before passing through a 70 μm cell strainer. Tumor-infiltrating leukocytes (TILs) were isolated using 67%:44% Percoll (Cytiva, Marlborough, MA, USA) gradient centrifugation. Single-cell suspensions from the spleens were prepared using mechanical disruption and passing through cell strainers followed by red blood cell lysis. In mice with mEER oral tumors, the cervical lymph nodes were collected, and in mice with TC-1 vaginal tumors, the inguinal lymph nodes were collected. In both cases, these tumor-draining lymph nodes were processed using mechanical disruption through a 70 μm cell strainer followed by centrifugation and re-suspension in complete RPMI medium.

### 2.7. Flow Cytometry

Following isolation, cells were incubated at 37 °C in 5% CO_2_ for 4–6 h with GolgiPlug/Brefeldin A (ThermoFisher, Waltham, MA, USA). Cells were blocked using mouse Fc-block (anti-CD16/32), stained for surface markers, and fixed and permeabilized using the intracellular/Foxp3 Fix-Perm reagent kit from eBioscience (ThermoFisher, Waltham, MA, USA); this was followed by staining for intracellular/functional markers (Appendix A). FACS data acquisition was performed on a five-laser Fortessa X-20 flow cytometer (BD Bioscience) and analyzed using FlowJo version 10 (FlowJo LLC, Ashland, OR, USA). Forward and side scatter parameters were used to set the singlet and leukocyte gates. The fixable viability stain 510 included in the surface antibody cocktail was used to gate-out dead cells and only analyze viable cells. HPV-16 E7 antigen-specific CD8 T cells were detected using a specific tetramer reagent that was part of the multiparametric panel. The APC-labeled H-2Db epitope E7_49–57_ (RAHYNIVTF)-peptide-loaded tetramer was obtained from the NIH MHC tetramer core facility at Emory University (Atlanta, GA, USA). For the detection of NKDCs, the gating strategy involving CD3-NK1.1+ CD11b+(int) followed by expression of CD11c+ (Appendix A) was followed, as previously described by Terme et al. [29].

### 2.8. Statistical Analysis

All graphing and statistical analyses were performed using GraphPad Prism version 8 (San Diego, CA, USA). An ordinary one-way ANOVA was used for comparing multiple groups, and a two-sample Student’s *t*-test was applied for comparing two groups. Statistical significance for matched pairs was determined using the Wilcoxon matched pairs test. Statistical significance between the survival curves was calculated using a log-rank test (Mantel–Cox). Graphs show the mean ± standard deviation (SD) for each group unless otherwise indicated. Experiments were repeated at least twice to confirm the reproducibility of results.

## 3. Results

### 3.1. Association of Innate Immunity from NKDCs with Vaccine-Mediated Protection in the Oral HPV Tumor Mouse Model

Based on the previously reported efficacy of the HPV-16 E6/E7 peptide therapeutic vaccine adjuvanted with aGalCer and CpG (TVAC) in the TC-1-Luc vaginal HPV tumor model [21], we tested intranasal administration of TVAC to mEER oral HPV tumor-bearing mice as described before [23,24]. Mice with visible tongue tumors by day 5 were randomized and either untreated or treated with TVAC or the therapeutic vaccine containing single adjuvants aGalCer (TVA) or CpG-ODN (TVC) by the intranasal route on days 5 and 11 after tumor implantation (Figure 1A). We observed that mice treated with TVAC and TVC as well as untreated groups experienced a high tumor burden and were sacrificed on day 51 (Figure 1B). Thus, in contrast to the significant TVAC-mediated tumor-free survival we reported in mice with vaginal HPV tumors [21], the present data indicate poor efficacy of TVAC against oral HPV tumors.

Because TVAC provided ineffective protection in the oral tumor model, we tested a modified therapeutic vaccine to incorporate the combination of QS21 and CpG-ODN (TVQC), where QS21 was used in place of aGalCer for intranasal administration in the mEER oral model. We observed that 72% of mice treated with TVQC showed sustained tumor regression and extended survival for 10 weeks following tumor challenge; this was significantly higher when compared to the control groups, including the untreated mice (Figure 1C). In addition, the tumor volume based on MRI measurements of the oral tumors on day 18 showed a significant reduction in tumor size in the TVQC-treated mice compared to the untreated mice (Figure 1D). Analysis of the tumor-infiltrating leukocytes (TILs) indicated that both TVAC and TVQC were effective in inducing strong HPV E7 antigen-specific CD8 T cell responses, even though the levels were higher in the mice treated with TVQC (Figure 1E). Thus, these data, while clearly demonstrating the enhanced effectiveness of TVQC relative to TVAC to protect against oral HPV tumors, the immunological responses beyond the adoptive HPV-specific CD8 T cells are unclear.

We therefore carried out multiparametric flow cytometric analyses of TILs from mice treated with the two vaccines in comparison to those in untreated mice, focusing on the levels of the innate immune NK cell response (Figure 2). We observed significantly elevated frequencies of total and functional (granzyme B- and/or IFNg-expressing) NK cells in mice treated with TVQC (Figure 2A). Importantly, a subset of total and functional NK cells that express the DC marker CD11c, referred to in the literature as NKDCs, were significantly higher in mice treated with TVQC compared to TVAC-treated mice (Figure 2B). To evaluate the contribution of NKDCs to the efficacy of TVQC, we compared the CD11c+ NK cells (NKDCs) to the CD11c− subset within the tumor microenvironment (TME) of TVQC-vaccinated mice. We observed significantly elevated frequencies of total as well as granzyme B- and/or IFNg-expressing NKDCs compared to the CD11c− subset (Figure 2C). Together, these data highlight the importance of innate effector immunity, specifically NKDCs, in the therapeutic efficacy of TVQC. The frequency of CD4 and CD8 T cells was also analyzed (Appendix A).

### 3.2. Both HPV E7-Specific and NKDC Responses Are Associated with TVQC-Mediated Protection against HPV Vaginal Tumors

We further evaluated whether TVQC, effective against oral HPV tumors in the mEER model, would also be useful to treat vaginal HPV tumors using the TC-1 model as an immunocompetent preclinical surrogate for HPV+ genital cancers [26,30]. Syngeneic C57BL/6 female mice were implanted with intravaginal TC-1luc tumor cells, as described in the Methods and Materials, and treated with TVQC delivered by the intranasal route on days 5 and 11 post-tumor implantation (Figure 3A). We observed tumor regression (Figure 3B) along with significant survival advantages in 80% of the vaccinated mice (Figure 3C). The therapeutic efficacy of TVQC correlated with significant induction of total and IFNg-producing tumor-antigen-specific CD8 T cells (Figure 3D) and NKDCs expressing granzyme B, IFNg or both (Figure 3E) in the tumor when compared to the untreated controls.

## 4. Discussion

Data presented here demonstrate that the adaptive immunity from total and HPV-antigen-specific CD8 T cells as well as the innate immunity from NK cell subsets both contribute towards the therapeutic vaccine-mediated protection against HPV oral tumors in the established preclinical model for HPV+ oropharyngeal cancers. Importantly, the vaccine containing the combination of QS21 and CpG-ODN as adjuvants (TVQC) was effective in inducing NKDCs, a novel subset of NK cells that express the dendritic cell marker CD11c, while the vaccine formulation with aGalCer in place of QS21 (TVAC) was less efficient in inducing NKDCs despite priming strong adaptive immunity via significant frequencies of antigen-specific CD8 T cells by both vaccines. Since robust induction of HPV-antigen-specific CTLs by TVAC was sufficient to treat TC-1 vaginal tumors [21], it appears that protection in the orthotopic mEER oral tumors requires induction of both an HPV-antigen-specific CTL response and a functional innate NK cell response in the form of functional NKDCs.

The NKDCs have been reported to possess both NK and antigen-presenting cell (APC) functions, potentially capable of cross-presenting tumor antigens [29], thus acting as a key link between the innate and adaptive immune responses. Even though a formal analyses for the APC function of NKDCs was not conducted in the present investigation, we believe that NKDCs contributed to the enhanced adaptive antigen-specific CTL response, as indicated by the relatively high frequency of HPV E7-specific CD8 T cells in the TME of mice treated with TVQC vs. those of TVAC (Figure 1E).

Innate, antigen-independent immune responses are key components of the tumor microenvironment, and their importance in promoting better cancer therapeutic outcomes is well known [9,10,31]. Moreover, in virus-associated cancers, such as the HPV+ cervical and oropharyngeal tumors, NK cells are known to be crucial in immunosurveillance and antitumor immunity [12]. In fact, the presence of activated NK cells in head and neck squamous cell carcinoma (HNSCC) has been shown to correlate with significantly better clinical outcomes [11,32]. However, we believe our study is the first report to show the role of NK responses in determining the efficacy of therapeutic vaccines against HPV-associated cancers, as well as the adequate induction of viral antigen-specific CD8 cytotoxic T cells (CTLs). The important attribute of TVQC efficacy in the current investigation is the robust induction of multiple effector immune responses, including polyfunctional NKDCs and tumor antigen-specific CTLs.

It is important to note that while total NK cells provide important contributions to the innate effector response of TVQC, in our studies, we observed that the CD11c+ NK subset (NKDC) exhibited a more potent functional phenotype than the CD11c− subset (Figure 2C), highlighting the significance of NKDCs in antitumor immunity. Similar to our findings, others have reported increased IFNg production by NKDCs compared to total NK cells [33,34]. Here, in addition to elevated IFNg, we show enhanced cytotoxicity shown by granzyme B and higher polyfunctionality (GrnzB+, IFNg+) from TVQC, all of which are known to be key components of the antitumor response [34,35].

The NK subset, NKDCs, has been reported as a potent innate immune responder with lytic capabilities and serves as an important source of IFNg production when activated [33,36]. However, a limitation of NKDCs in the immune response is their low frequency in circulation [33]. The scarcity of NKDCs highlights the importance of identifying a means to expand and activate them. It is known that adjuvants, as essential components of vaccine formulations, play an important role in mediating the production of cytokines, which in turn promotes various immune effector populations. Published reports show that aGalCer induces IL-4 and IFNg; CpG-ODN induces type I IFN; and QS21 induces IL-1, IL-18, and IL-12 [37,38,39,40,41,42]. The precise mechanisms underlying the effectiveness of the QS21 and CpG-ODN adjuvant combination to promote significant innate and adaptive cellular immune responses observed in the present investigation are unclear. Semmling et al. [43,44] described the phenomenon termed the dual licensing of DCs. They used a combination of adjuvants that use different signaling mechanisms to induce the secretion of distinct sets of chemokines to recruit multiple sets of effector immune cells. It is possible that the balanced Th1 and Th2 responses from QS21 [18], when combined with TLR9-mediated signaling from the CpG-ODN in the TVQC vaccines, promoted the effective induction of multiple effector immune responses. It is known that IFNg is important in antitumor immunity by promoting NK and CD8 CTL functions, while IL-15 modulates antitumor immunity by promoting NK cells and CD8 T cell memory [45,46,47,48]. Moreover, IL-15 by itself has been used as a mucosal adjuvant along with the BCG vaccine and autologous whole-cell lung tumor vaccines in preclinical models [49,50,51]. Chaudry et al. reported that expansion with IL-15 significantly enhanced IFNg production and proliferation of NKDCs [33]. Furthermore, combinations of IL-2, IL-15, IL-18, and IL-21 have been used to successfully expand the total and subsets of NK cells [52,53], and we obtained preliminary data showing higher plasma levels of these cytokines in mice treated with TVQC . Overall, the elevated functionality and expansion potential of NKDCs, by cytokines or adjuvants, makes them an attractive subset for cellular immunotherapy to promote effective antitumor immunity.

## 5. Conclusions

Overall, evidence from the present investigation into preclinical oral and vaginal HPV tumor models supports and extends our earlier observations concerning the effectiveness of therapeutic HPV peptide vaccination by incorporating combinations of diverse adjuvants. While adaptive immunity, in terms of antigen-specific CD8 T cells, was essential for protection in both the preclinical models, the present investigation highlights the importance of the innate immunity afforded by NKDCs, a subset of NK cells, in vaccine-mediated protection against oral HPV tumors. Importantly, incorporation of QS21, a saponin adjuvant, along with CpG-ODN into the vaccine formulation (TVQC) was effective in inducing the NKDC responses. In TVQC-vaccinated mice, NKDCs in the TME exhibited high polyfunctionality, indicative of high levels of cytokine secretion and cytotoxicity. Importantly, we observed significant functional advantages in the CD11c+ (NKDC) vs. the CD11c− subset, highlighting the importance of this NK subset in the antitumor immunity mediated by the vaccine. Future studies testing the depletion and/or adoptive transfer of NKDCs should further substantiate their protective role. Additionally, since NKDCs express the DC marker CD11c, it would be interesting to investigate their potential contributions as antigen-presenting cells.

## Figures and Tables

**Figure 1 vaccines-12-00206-f001:**
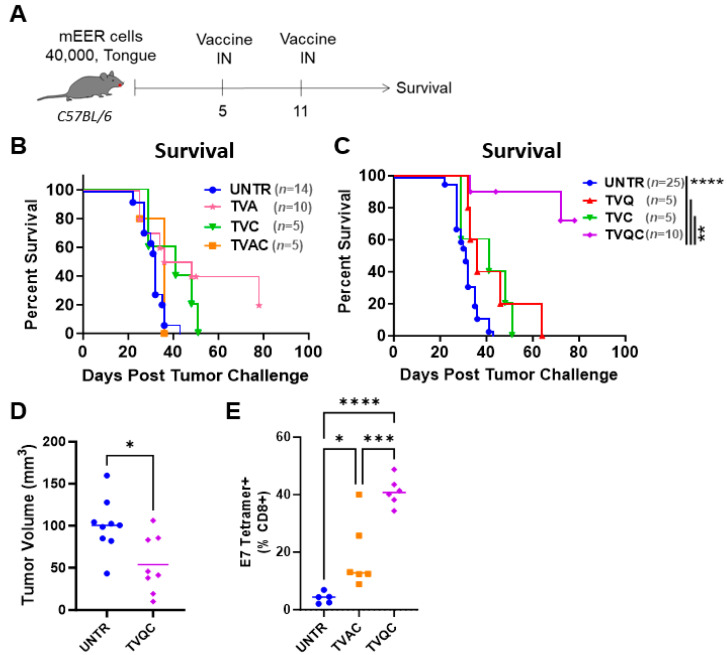
Differential efficacy of TVAC vs. TVQC in the oral HPV tumor model. Syngeneic C57Bl/6 mice were injected with mEER tumor cells as indicated and treated with intranasal (IN) delivery of the indicated E6/E7 HPV peptide therapeutic vaccine formulations on days 5 and 11 post-tumor implantation (**A**). Untreated mice (UNTR) and mice vaccinated with single adjuvants (TVA, TVC, TVQ) served as controls. Mice in all the groups were monitored over time, and Kaplan–Meier survival curves are shown for mice treated with TVAC (**B**) and TVQC (**C**). The Mantel–Cox log-rank test, ** *p* < 0.005, **** *p* < 0.0001 (*n* = 5–25 mice per group), shows a significant regression of mEER oral tumors with the TVQC vaccine. The oral tumor volume for mice in the untreated and TVQC-treated groups, as determined using magnetic resonance imaging (MRI) analyses on day 18, showed a significant reduction in tumor size in vaccinated mice (**D**). Elevated frequencies of antigen-specific CD8 T cells were observed in the tumors with both vaccine formulations (**E**). Significance was determined using an ordinary one-way ANOVA * *p* < 0.05, *** *p* < 0.0005, **** *p* < 0.0001 (*n* = 5–6 mice per group).

**Figure 2 vaccines-12-00206-f002:**
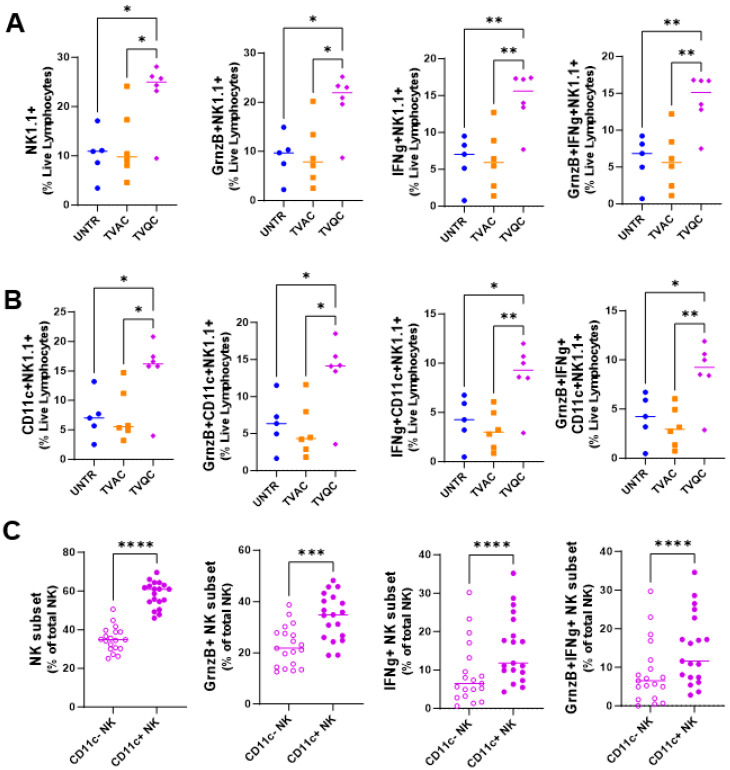
TVQC induces a more robust innate effector response than TVAC in mice with oral HPV tumors. Syngeneic C57Bl/6 mice were injected with mEER tumor cells as indicated and treated with intranasal (IN) delivery of TVAC, TVQC or untreated (UNTR). TVQC induced significantly elevated frequencies of total and functional (granzyme B- and/or IFNg-expressing) NK cells (**A**) and NKDCs (**B**) in the TME. Significance was determined using an ordinary one-way ANOVA * *p* < 0.05, ** *p* < 0.005 (*n* = 5–6 mice per group). Within the TVQC-vaccinated mice, the total and functional CD11c−/+ NK cells were compared with the CD11c+ subset (NKDCs) exhibiting a higher frequency and functionality in the TME (**C**). Significance was determined using a Wilcoxon matched pairs test **** *p* < 0.0001, *** *p* < 0.0001 (*n* = 19 mice).

**Figure 3 vaccines-12-00206-f003:**
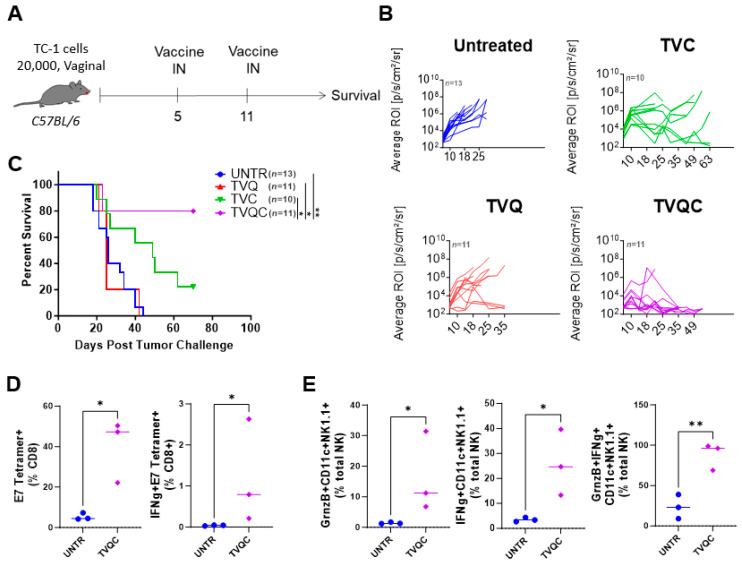
Syngeneic C57BL/6 female mice were implanted with intravaginal TC-1luc tumors, as described in the Materials and Methods, and treated with intranasal HPV peptide therapeutic vaccine formulations as indicated on days 5 and 11 post-tumor implantation (**A**). Intravaginal TC-1 tumor size was monitored by luciferase imaging using the IVIS bioluminescence imaging system, and tumor growth curves for the indicated groups are shown (**B**). Survival curves are shown for intravaginal TC-1 tumor-bearing mice as indicated (**C**). A Mantel–Cox log-rank test was used to determine the significance between the groups, * *p* < 0.05, ** *p* < 0.005 (*n* = 10–13 mice per group). Immune responses associated with vaccine efficacy were assessed by flow cytometric analysis of TILs on day 16 following tumor implantation. The frequencies of the total and IFNg+ E7 tetramer+ CD8 T cells (**D**) and NKDCs expressing granzyme B, IFNg or both are shown (**E**). Significance was determined using an unpaired *t*-test, * *p* < 0.05, ** *p* < 0.01 (representative data from one experiment with *n* = 3 mice per group; the experiment was repeated one more time).

## Data Availability

The raw data supporting the conclusions of this article will be made available by the authors on request.

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
