# Peer review of "Immunity from NK Cell Subsets Is Important for Vaccine-Mediated Protection in HPV+ Cancers"

_vaccines, 2024, doi:10.3390/vaccines12020206_

Round 1

Reviewer 1 Report

Comments and Suggestions for Authors

I read the interesting paper by O'Hara and colleagues on the role of NK cells for protection of HPV+ cancers. Here are my comments:

- In section 2.4, please add the precise number of tumor bearing mice that were used in each group for each experiment;

- In section 2.5, please provide a better description of TVAC and TVQC vaccine composition and of all adjuvant/oncopeptides included. Use a table to correlate vaccine modifications and treated mouse models;

- In section 2.6, please describe better the digestion + single cell suspension protocol of the tumor draining lymph nodes. Which lymph nodes exactly did you extract for each tumor model?

- In section 3.1 please do not repeat the model description, as it is already included in section 2.4. 

- Discussion present a nice collection and description of findings on NKDC cells, adjuvants, role of innate and adaptive immune response for tumor protection. Please, connect better this literature with 1) the in vivo results you obtained, especially those of TC1-luc tumor model, and 2) with the role of the adjuvants you used.

In addition, please add a future prospective in order to obtain a more mechanistic explanation of the results.

Reviewer 2 Report

Comments and Suggestions for Authors

In this study, the authors used preclinical HPV tumor mouse model, and intranasal immunization of HPV peptide therapeutic vaccine to assess the role of NK and CD8+ cells in immunity. The authors reported that that the induction of significantly higher over all innate NK effector responses by TVQC relative to TVAC.

Major comments

1) The authors need to show the number of mice/ group and replace the figure with dots to show the numbe rof mice/group.

2) It is not clear that NK cell and CD8 cell isolated from tumor, spleen or lymph nodes or mix.

3) Please show the pecentage of CD4+ cells. CD4+/CD8+ ratio, in the mice model.

4) Can the authors provide the tumor volume in each group.

5) DId the author measure IFNg and other cytokines in the plasma of mice.

6) Figure 1, it is not clear the difference between 1B and 1C and the home message from these sub-figures.

Comments on the Quality of English Language

moderate language editing

Reviewer 3 Report

Comments and Suggestions for Authors

Dear Authors

The manuscript is well written, clear concise and with a good rationale behind it.

Title should be mor direct to the study.

The introduction is very good and more than enough to make the context and identify the gap.

Although the authors do not finish the introduction with the aim of the work, please rewrite from lines 62 to 69.

Material and methods

All reagents and all equipment must have between brackets the producer, city and country.

All abbreviations must first be written fully. examples DMEM, RPMI, FBS.

Abbreviations do not have plural, please correct TILs must be TIL, CTLs must be CTLcheck all manuscript.

versus is Latim must be in italic.

A table with the antibodies used would facilitate the reading of the manuscript.

Missing the study limitations and conclusions.

Did the authors base all results only on two independent experiments? preferably 3 independent experiments, n=3.

Round 2

Reviewer 2 Report

Comments and Suggestions for Authors

In the revised manuscript, the authors did not provide satisfactory responses to my comments.

Besides, two major points raised that could affect the quality of data and final conclusion

a) The number of mice (as shown by dots) are not the same in the whole figures, this concern could raise the bias in selection of data and interpretation.

b) some data in figures 2 the panels look identical, this point need evaluation of data quality from the authors

Comments on the Quality of English Language

Moderate language editing

Round 3

Reviewer 2 Report

Comments and Suggestions for Authors

No further comments

Comments on the Quality of English Language

Moderate language editing